# Comparative Proteomic Analysis of Transcriptional and Regulatory Proteins Abundances in *S. lividans* and *S. coelicolor* Suggests a Link between Various Stresses and Antibiotic Production

**DOI:** 10.3390/ijms232314792

**Published:** 2022-11-26

**Authors:** Lejeune Clara, Cornu David, Sago Laila, Redeker Virginie, Virolle Marie-Joelle

**Affiliations:** 1Institute for Integrative Biology of the Cell (I2BC), Department of Microbiology, Group “Energetic Metabolism of Streptomyces”, CEA, CNRS, Université Paris-Saclay, 91198 Gif-sur-Yvette, France; 2Laboratory of Neurodegenerative Diseases, Commissariat à l’Energie Atomique et aux Energies Alternatives (CEA) and Centre National de la Recherche Scientifique (CNRS), Molecular Imaging Center (MIRCen), Institut François Jacob, Université Paris-Saclay, 92260 Fontenay-aux-Roses, France

**Keywords:** *Streptomyces*, antibiotics, nitrogen stress, cell wall stress, osmotic stress, proteomics, label-free protein quantification

## Abstract

*Streptomyces coelicolor* and *Streptomyces lividans* constitute model strains to study the regulation of antibiotics biosynthesis in *Streptomyces* species since these closely related strains possess the same pathways directing the biosynthesis of various antibiotics but only *S. coelicolor* produces them. To get a better understanding of the origin of the contrasted abilities of these strains to produce bioactive specialized metabolites, these strains were grown in conditions of phosphate limitation or proficiency and a comparative analysis of their transcriptional/regulatory proteins was carried out. The abundance of the vast majority of the 355 proteins detected greatly differed between these two strains and responded differently to phosphate availability. This study confirmed, consistently with previous studies, that *S. coelicolor* suffers from nitrogen stress. This stress likely triggers the degradation of the nitrogen-rich peptidoglycan cell wall in order to recycle nitrogen present in its constituents, resulting in cell wall stress. When an altered cell wall is unable to fulfill its osmo-protective function, the bacteria also suffer from osmotic stress. This study thus revealed that these three stresses are intimately linked in *S. coelicolor*. The aggravation of these stresses leading to an increase of antibiotic biosynthesis, the connection between these stresses, and antibiotic production are discussed.

## 1. Introduction

Actinomycetes, and especially *Streptomyces*, are industrially and medically important Gram-positive bacteria responsible for the production of the majority of antibiotics in current use [1]. Nutritional limitations in phosphate (Pi) and/or nitrogen (N) are thought to trigger a complex morphological differentiation process that usually coincides with the production of specialized bio-active metabolites [2,3]. However, the apprehension of the nature of the extra- and/or intra-cellular signals [4] and of the role of the global and specific regulators, and their interplay [5], in the triggering of the biosynthesis of specialized metabolites is still incomplete, even after half a century of active research. *Streptomyces coelicolor* (*SC*) [6] and *Streptomyces lividans* (*SL*) [7] are the model strains extensively used to study the regulation of specialized metabolites biosynthesis since these closely related strains possess the same biosynthetic pathways directing the biosynthesis of three well-characterized bio-active specialized metabolites, calcium-dependent antibiotic (CDA), undecylprodigiosin (RED), and actinorhodin (ACT). However, these metabolites are mainly produced by *SC* [8] and, consistently, proteins encoded by genes of these pathways are far more abundant in *SC* than in *SL* [9]. In order to gain a better understanding of the different physiological and metabolic features underpinning the contrasted biosynthetic abilities of these two strains to produce antibiotics and of the impact of phosphate availability on the latter, an in-depth comparative analysis of the proteome of these two strains grown for 48 and 60 h on the solid modified R2YE medium either limited (1 mM) or proficient in Pi (5 mM), was carried out. Relative abundance and regulatory features of proteins of the primary and specialized metabolisms of these strains were previously published in Lejeune et al. [9]. In this issue we present a comparative analysis of the abundance of proteins of the transcriptional apparatus and regulatory proteins of one and two component systems, as well as eukaryotic-like protein kinases of these two strains. This report is based on the same set of proteomic data as Lejeune et al. [9], so efforts were made throughout the present manuscript to establish correlations between the abundance of mentioned regulators and that of their established or putative regulatory targets via the systematic quotation of the figure numbers of Lejeune et al. [9].

The study of Lejeune et al. [9] revealed major differences between *SC* and *SL*. One was the far higher abundance of subunits of the complex I/NADH dehydrogenase of the respiratory chain, (nuoABCDEFGHIJKLMN/SCO4562-75) in *SC* compared to *SL* that might contribute to the reported high ATP content of this strain [10] as well as to the generation of ROS/NOS [11] leading to oxidative stress (OxS), a proposed important trigger of ACT biosynthesis [8,12]. The other was the poor abundance of proteins involved in nitrogen up-take and assimilation in *SC* compared to *SL*, likely to result in severe nitrogen stress. Since nitrogen (N), carbon (C), and phosphate (P) assimilation ought to be regulated in *Streptomyces* [13], as in all living organisms [14], the poor ability of *SC* to assimilate N is correlated with the reduced abundance of proteins playing a role in C up-take and assimilation such as the glucose permease (SCO5578) (Figure 16B of [9]) and all glycolytic enzymes (Figure 2 of [9]) in *SC* compared to *SL*. Similarly, proteins of the Pho regulon involved in Pi up-take and assimilation were less abundant in *SC* than in *SL* but not to the same extent as those of N and C metabolisms [8].

The present study indicated that the abundance of the vast majority of proteins of the transcriptional apparatus, of one or two component transcriptional regulators, and of regulatory proteins such as eukaryotic-like serine/threonine protein kinases, common to *SC* and *SL*, greatly differed between the two strains and responded differently to Pi availability.

The abundance of specific transcriptional and regulatory proteins revealed in this issue was consistent with the existence of nitrogen stress in *SC* and also demonstrated the existence of cell wall and osmotic stress in this strain. We thus proposed that, in *SC*, nitrogen stress would trigger the degradation of the nitrogen-rich peptidoglycan cell wall in order to recycle nitrogen present in its constituents. The alteration of the cell wall would result in cell wall stress. Since the alteration of the cell wall likely hinders its osmo-protective function, the bacteria would also suffer from osmotic stress. This study thus indicated that nitrogen, cell wall, and osmotic stresses are intimately linked in *SC* and that nitrogen stress is likely to be the original cause of the other stresses. A plausible cause of the origin of nitrogen stress in *SC* is addressed in the discussion. Furthermore, the higher abundance of transcriptional proteins positively controlling the expression of genes involved in the resistance to cell wall and osmotic stresses in *SC* compared to *SL*, indicates attempts of *SC* to maintain its cellular homeostasis. The reported correlation between the aggravation of these stresses and the increase of antibiotic biosynthesis suggests a link between these stresses and antibiotic production that is discussed.

## 2. Results

### 2.1. Components of the Transcriptional Apparatus

Figure 1 indicates that the abundance pattern of the 39 proteins of the transcriptional apparatus is utterly different between *SL* and *SC* and fall into four major clusters (a to d). Proteins of cluster a were more abundant in Pi limitation than in Pi proficiency in *SC* as well as in *SL* to a lesser extent; proteins of cluster b were more abundant in *SC* than in *SL* in both Pi conditions, whereas it was the opposite for proteins of cluster c, and proteins of cluster d were more abundant in Pi proficiency than in Pi limitation in both strains.

Cluster a includes 12 proteins rather more abundant in *SC* than in *SL* in Pi limitation especially at 60 h. This cluster includes the principal sigma factor HrdB/SCO5820 [15,16] [17], two subunits of the core RNA polymerase, RpoA/SCO4729 [18] and RpoC/SCO4655 [19]; the sigma factors SCO3613 and SigR that is a major regulator positively controling the expression of numerous genes involved in resistance to oxidative stress (OxS) [20,21,22,23,24], the anti-sigma factor SCO5244 and the anti-sigma factor antagonist BldG [25] are involved in the regulation of the activity of the osmotic stress (OsS) responsive sigma factor H [26,27], as well as the RNA polymerase-binding protein Dksa-like SCO6164. Proteins of this family are thought to stabilize ppGpp binding to RNA polymerase [28,29] but the function of SCO6164 is still unclear in Streptomyces. Four proteins of cluster A were also abundant in *SL* in Pi proficiency at both time points. It includes the ECF sigma factor BldN/SCO3323 required for aerial mycelium formation [30,31]; RpoZ/SCO1478 [32] that was shown to play a positive role in the regulation of morphological differentiation and antibiotic production [33]; the anti-termination protein NusG/SCO4647 [34], and the RNA nucleotidyltransferase SCO3896 [35].

**Figure 1 ijms-23-14792-f001:**
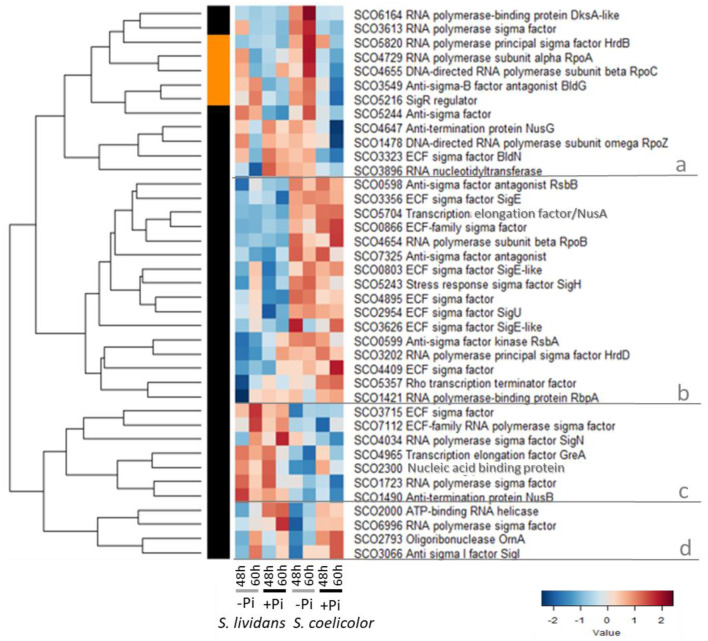
Heatmap representation of the abundance of components of the transcriptional apparatus of *SL* and *SC* that can be divided in 4 sub-clusters a, b, c and d. Cluster a includes 12 proteins rather more abundant in *SC* than in *SL* in Pi limitation especially at 60 h. Cluster b includes 16 proteins more abundant in *SC* than in *SL* in both Pi conditions. Cluster c includes 7 proteins more abundant in *SL* than in *SC* in both Pi conditions. Cluster d includes 4 proteins more abundant in Pi proficiency than in Pi limitation in both strains but mainly at 60 h.

Cluster b includes 16 proteins more abundant in *SC* than in *SL* in both Pi conditions. This cluster includes the β subunit RpoB/SCO4654 of the core RNA polymerase able to interact with the stringent factor ppGpp [36], the RNA binding protein RbpA/SCO1421 that promotes the binding of the vegetative sigma factor to the core RNA polymerase [37], the transcription termination factor Rho/SCO5357 [38], and the transcription elongation factor SCO5704/NusA. The principal sigma factor HrdD/SCO3202 might be involved in the response to various stresses [39] and SigH/SCO5243 is involved in the response to OsS. It directs the transcription of the gene encoding the ECF sigma factor SigJ [40] and plays a positive role in the regulation of morphological differentiation and ACT production [41,42,43]. Cluster b also includes seven ECF-like sigma factors [44,45] such as SigE/SCO3356 and SigU/SCO2954, as well as SCO0803, SCO0866, SCO3626, SCO4409, and SCO4895. SigE belongs to a signal transduction system sensing and responding to general cell wall stress and is required for normal cell wall structure [46,47]. SigE contributes to the regulation of HrdD expression [48] and plays a negative role in the regulation of ACT biosynthesis [47,49]. SigU positively controls the expression of numerous genes encoding secreted proteins including proteases that may be important to combat cell wall stress [50] and it has a negative impact on morphological differentiation [51]. The pair anti-sigma factor kinase RsbA/SCO0599-anti-sigma factor antagonist RsbB/SCO0598 [52,53], and the anti-sigma factor antagonist RsbV/SCO7325 [52] are also present in this cluster. These proteins are thought to control the activity of sigma B/SCO0600, a sigma factor playing a role in osmoprotection and differentiation [53] not detected in our study.

Cluster c includes 7 proteins more abundant in *SL* than in *SC* in both Pi conditions. This cluster includes sigma factors SCO1723 and SigN/SCO4034 that are involved in general stress response and have a positive impact on antibiotic production and morphological differentiation [54,55] as well as two ECF sigma factors (SCO3715 and SCO7112). It also includes the elongation factor GreA-like/SCO4945, the antitermination protein NusB/SCO1490, and the nucleic acid binding protein SCO2300.

Cluster d includes 4 proteins more abundant in Pi proficiency than in Pi limitation in both strains but mainly at 60 h. These include the anti-sigma I factor/SCO3066/PrsT that together with the anti-sigma I factor antagonist ArsI/SCO3067 regulates the activity of SigI/SCO3068 whose expression is induced by OsS [56]. This cluster also includes the sigma factor SCO6996, the oligoribonuclease OrnA/SCO2793 necessary for active vegetative growth and morphological differentiation [57], and an ATP-binding RNA helicase/SCO2000.

These data revealed that 71% of the proteins of transcriptional apparatus detected, including eight ECF sigma factors, were more abundant in *SC* than in *SL* in both Pi conditions (16 proteins of cluster B) or mainly in Pi limitation (12 proteins of cluster A). Many of these proteins were shown to be involved in the resistance to various stresses such as cell wall stress (SigE, SigU), osmotic stress (sigH, sigB), and oxidative stress (SigR) and to have a negative impact on antibiotic production and morphological differentiation. This indicated that *SC*, in contrast to *SL*, “suffers” multiple stresses and this raises the question of the origin of these stresses in *SC* (see Discussion).

### 2.2. Transcriptional Regulators and Sensory Histidine Kinases

Figure 2, Figure 3, Figure 4, Figure 5, Figure 6, Figure 7, Figure 8 and Figure 9 indicate that the abundance patterns of the 293 transcriptional regulators and sensory histidine kinases fall into eight major clusters (A to H). Some were more abundant in Pi proficiency than in Pi limitation (Figure 2; cluster A) or more abundant in Pi limitation than in Pi proficiency (Figure 3, cluster B) in both strains. Other regulators were more abundant in *SC* than in *SL* in both Pi conditions (Figure 4, cluster C) or mainly in Pi proficiency (Figure 5, cluster D) or mainly in Pi limitation (Figure 6, cluster E) when other regulators were more abundant in *SL* than in *SC* in both Pi conditions (Figure 7, cluster F) or mainly in Pi limitation (Figure 8, cluster G) or mainly in Pi proficiency (Figure 9, cluster H).

#### 2.2.1. Cluster A: Regulators More Abundant in Pi Proficiency than in Pi Limitation in Both Strains

This cluster could be divided in three sub-clusters, A1, A2, and A3 (Figure 2).

Sub-cluster A1 includes 22 regulators rather more abundant in Pi proficiency than in Pi limitation in both strains. Some of these regulators were more abundant in *SL* than in *SC* in Pi proficiency, such as the HU DNA-binding protein SCO5556 [58] whose poor abundance in *SC* is likely to have a global impact on gene expression. Other regulators were more abundant in *SC* than in *SL* in both Pi conditions, such as ScbR/SCO6265 which positively controls the expression of the γ-butyrolactone synthase ScbA/SCO6256 [59,60]. ScbR in interaction with ScbA is involved in a bi-stable switch governing antibiotic production [61,62].

However, most regulators had a similar abundance in the two strains, such as TamR/SCO3133 whose expression is induced in condition of OxS [63], the heat-inducible repressor HrcA/SCO2555 that controls positively the transcription of the groES-groEL1 operon, and of the groEL2 gene [64,65] and sco7639 that is located in divergence of the enolase encoding gene sco7638. The regulator SCO7639 and the enolase SCO7638 were clearly more abundant in Pi proficiency than in Pi limitation in *SL* (Figure 2 of [9]) suggesting that SCO7639 positively regulates sco7638 expression in conditions of Pi proficiency. However, despite the similar abundance of SCO7639 in *SC* and in *SL*, the enolases SCO7638 and SCO3096 were poorly abundant in *SC* (Figure 2 of [9]). This suggested that the presence/absence of a molecule acting as an allosteric effector or that a post-translational modification impairs the regulatory function of SCO7639 in *SC*.

The sub-clusters A2 and A3 both include 16 regulators that were similarly abundant in *SL* and *SC* in Pi proficiency at both time points but also abundant in *SL* in Pi limitation at 60 h. Sub-cluster A2 includes:-The response regulator (RR) OsaB/SCO5749 of TCS OsaA (HK)/OsaB whose expression is induced in conditions of OsS in a sigma B-dependent manner [66]. An OsaB mutant cannot erect aerial hyphae and produces up to 5-fold more antibiotic than the wild-type strain [66]. This TCS, involved in osmoadaptation, thus plays a negative role in the regulation of antibiotic production suggesting that high OsS contributes to the triggering of antibiotic production.-The sensory histidine kinase (SHK) DraK/SCO3062 of the TCS DraK/DraR. DraR was shown to directly activate the transcription of pathway-specific activator actII-ORF4 and thus ACT biosynthesis [67] whereas it represses CPK and RED biosynthesis via the repression it exerts on the expression of KasO, the specific activator of the CPK pathway, but its effect on RED biosynthesis is independent of the pathway-specific activators RedD and RedZ [67].-The regulator DmdR1/SCO4394 controls iron homeostasis via the repression it exerts on the expression of the *desABCDE/sco2782-85* cluster that directs the biosynthesis of the siderophore, desferrioxamine. Such repression is relieved in conditions of iron deprivation. Our previous data indicated that DesABCE were more abundant in *SL* than in *SC*, especially in Pi proficiency, whereas, unexpectedly, the siderophore biosynthetic enzyme DesD, was highly abundant in both Pi conditions in *SC* (Figure S12B of [9]).-The SOS regulatory protein LexA/SCO5803 that positively controls the expression of the recombination protein RecA/SCO5769 [68]. Consistently, LexA and RecA were more abundant in Pi proficiency than in Pi limitation in *SL* but RecA was similarly and highly abundant in both Pi conditions in *SC* (Figure S8 of [9]) indicating a situation of stress.-The gene encoding the regulator SCO1614 that is located in divergence of an ethanolamine utilization pathway [69].

**Figure 2 ijms-23-14792-f002:**
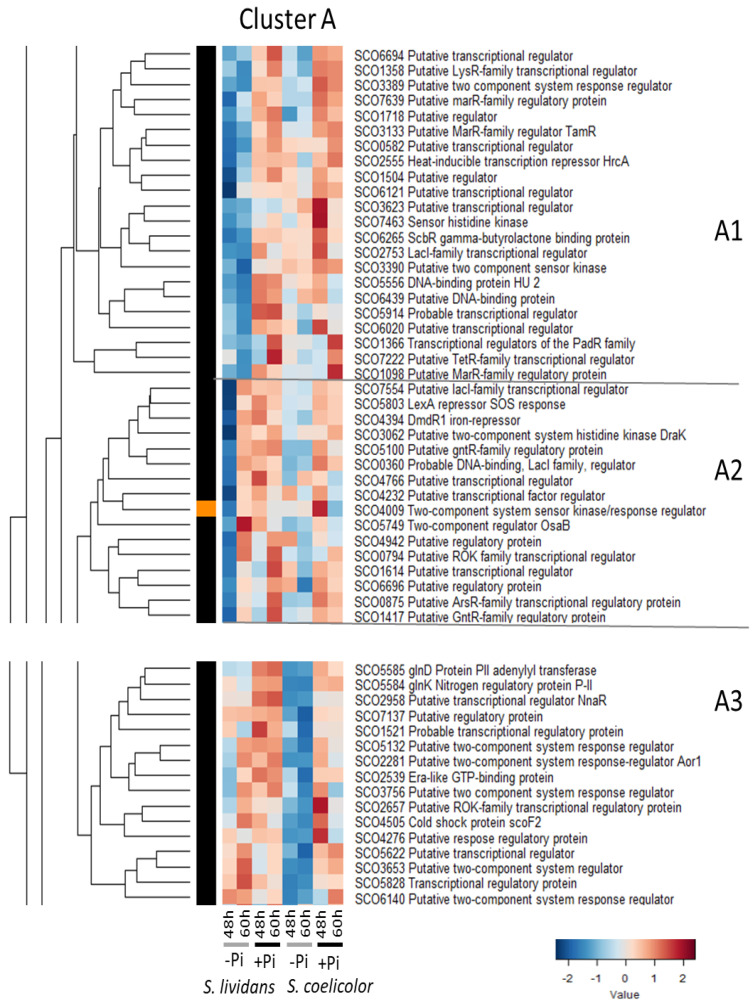
Cluster A, heatmap representation of transcriptional regulators and sensory histidine kinases more abundant in Pi proficiency than in Pi limitation in both strains. Cluster A that can be divided into 3 sub-clusters A1, A2 and A3. Sub-cluster A1 includes 22 regulators rather more abundant in Pi proficiency than in Pi limitation in both strains. The sub-clusters A2 and A3 both include 16 regulators that were similarly abundant in *SL* and *SC* in Pi proficiency at both time points but also abundant in *SL* in Pi limitation at 60 h.

Sub-cluster A3 includes 3 proteins involved into the regulation of nitrogen metabolism: the regulator SCO2958/nnaR, the adenylyltransferase GlnD/SCO5585, and the nitrogen sensor protein GlnK/SCO5584. NnaR is a regulatory target of the nitrogen regulator GlnR/SCO4159 and acts as a co-activator of the latter [70]. NnaR directly positively controls the expression of the genes narK (nitrate extrusion protein, SCO2959), nirB (nitrite reductase, SCO2486-87), nirA (nitrite/sulphite reductase SCO6102), and nasA (nitrate reductase SCO2473) that are essential for nitrate and nitrite assimilation [70]. NnaR was more abundant in Pi proficiency than in Pi limitation in both species, but NnaR was far less abundant in *SC* than in *SL* in both Pi conditions. Consistently all the NnaR target proteins were far more abundant in Pi proficiency than in Pi limitation in *SL* but not in *SC* (Figure 7 of [9]). GlnD adenylates GlnK/SC05584 [71] which has a positive impact on antibiotic production and morphological differentiation [72]. The low abundance of all these proteins in Pi proficiency in *SC* likely limits its ability to assimilate nitrate/nitrite and should result in severe nitrogen limitation in *SC*.

This cluster also includes the orphan RR SCO2281/Aor1 that has a positive impact on antibiotic biosynthesis and morphological differentiation [73]. In an aor1 mutant of *SC*, genes of the SigB regulon were up-regulated, suggesting a link with OsS [53,74].

#### 2.2.2. Cluster B: Regulatory Proteins More Abundant in Pi Limitation than in Pi Proficiency in Both Strains

Cluster B could be divided into three sub-clusters: B1, B2, and B3 (Figure 3).

Sub-cluster B1 includes 6 proteins more abundant in *SC* than in *SL* in both Pi conditions but also abundant in *SL* in Pi limitation at 48 h. Sub-cluster B1 includes:-The SHK CseC/SCO3359 of the TCS *cseC/cseB* belongs to an operon constituted also by *cseA* (lipoprotein) and *sigE*. This TCS activates the transcription of *sigE/sco3356*, in response to the alteration of the cell envelope [47,75]. Consistently, SigE was far more abundant in *SC* than in *SL* in both Pi conditions (Figure 1). This indicated that damages to the cell wall occur independently of Pi availability in *SC*.-AbrC1/SCO4598 is part of an atypical TCS constituted of 2 SHK (AbrC1/SCO4598 and AbrC2/SCO4597) and of the RR AbrC3/SCO4596 [76]. The genes encoding this TCS are located in divergence of genes encoding sub-units of NADH dehydrogenase (SCO4599-SCO4608) which were far more abundant in *SC* than in *SL* in both Pi conditions (Figure 5 of [9]). This TCS has a positive impact on antibiotic production and morphological differentiation in *SC* [76,77,78].-The regulator BldB/SCO5723 that has a positive impact on antibiotic production and morphological differentiation [79].

Sub-cluster B2 includes 22 proteins clearly more abundant in Pi limitation than in Pi proficiency in both strains. Sub-cluster B2 includes:-Three proteins involved in the regulation of phosphate metabolism: the TCS PhoR/PhoP as well as the regulatory protein PhoU/SCO4228. In condition of Pi limitation, PhoR/PhoP positively controls the expression of genes of the Pho regulon involved in Pi scavenging and up-take [80] and negatively those involved in nitrogen assimilation [81]. This TCS also plays a negative role in the control of morphological differentiation and antibiotic production [82,83]. PhoU, through its interaction with PstB, the ATP-binding cassette of the high affinity ABC Pi transporter PstSCAB, was proposed to sense environmental Pi and to transmit this signal to the SHK PhoR, promoting or inhibiting its auto-phosphorylation in Pi limitation or proficiency, respectively [84,85]. The difference in the abundance of proteins involved in Pi scavenging and up-take, in conditions of in Pi limitation or proficiency, was much greater in *SL* than in *SC* (Figure 6 of [9]).-The adenylyltransferase GlnE/SCO2234 that modulates the activity of the glutamine synthase GSI through adenylylation, in response to N availability [86] and is thus involved in the regulation of nitrogen metabolism.-The LacI regulator SCO4158 [87] whose encoding gene is located downstream of *glnR/sco4159* encoding a major regulator of N metabolism (cluster G). SCO4158 was abundant in both strains in Pi limitation whereas GlnR was far more abundant in Pi proficiency than in Pi limitation in *SL* but poorly abundant in *SC* in both Pi conditions. Considering the proximity of *sco4158* and *glnR* and their opposite regulatory features, SCO4158 might regulate *glnR* expression.

Proteins involved in N assimilation were clearly more abundant, but only slightly more abundant, in Pi proficiency than in Pi limitation, in *SL* and *SC*, respectively (Figures 6 and 7 of [9]). This suggested that, besides the relieve of repression of the expression of these genes by PhoP in condition of Pi proficiency, an activator of the expression of these genes might be either missing or non-functional in *SC* (see Discussion).

**Figure 3 ijms-23-14792-f003:**
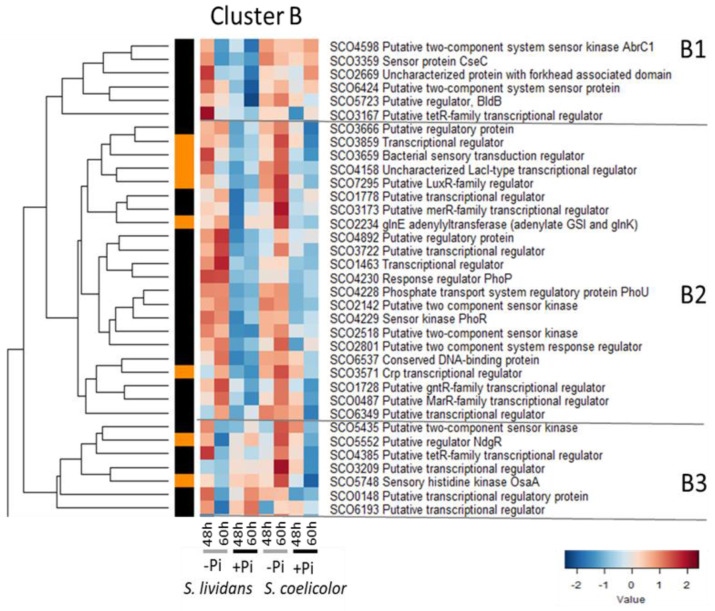
Cluster B, heatmap representation of transcriptional regulators and sensory histidine kinases more abundant in Pi limitation than in Pi proficiency in both strains. Cluster B that can be divided into 3 sub-clusters B1, B2 and B3. Sub-cluster B1 includes 6 proteins more abundant in *SC* than in *SL* in both Pi conditions but also abundant in *SL* in Pi limitation at 48h. Sub-cluster B2 includes 22 proteins more abundant in Pi limitation than in Pi proficiency in both strains. Sub-cluster B3 includes 7 regulators more abundant in Pi limitation than in Pi proficiency at 48 h and 60 h in *SL* and *SC*, respectively, and more abundant in *SL* than in *SC* in Pi proficiency.


-The pleiotropic regulator Crp/SCO3571 [88,89] and the regulator AtrA/SCO4118 [90] both have a positive impact on ACT biosynthesis via their direct interaction with the promoter region of *actII-ORF4*.-The gene encoding the GntR regulator SCO1728 is located in divergence of a gene encoding a putative mycothiol synthase that might play a role in the resistance to OxS. In the condition of over-expression, SCO1728 was shown to have a negative impact on antibiotic production [91], which might be linked to its role in the resistance to OxS that was proposed to be an important trigger of ACT biosynthesis [8,12].-The MarR regulator SCO0487/CchL, whose encoding gene is present in the biosynthetic pathway of the peptide siderophore, coelichelin (*sco0499*/*cchA—sco0489*/*cchK*) [92]. The 11 proteins of the coelichelin cluster detected (Figure S12B of [9]) were, as SCO0487, all more abundant in Pi limitation than in Pi proficiency in *SL*, suggesting that SCO0487 positively regulates their expression in Pi limitation. Five of these proteins (SCO0493 to SCO0497) were also more abundant in Pi limitation than in Pi proficiency in *SC* whereas, unexpectedly, four others (SCO0492, SCO0498 to SCO0490) were more abundant in Pi proficiency than in Pi limitation in *SC* and two (SCO0499 and SCO0491) were poorly abundant in both Pi conditions in *SC*.


Sub-cluster B3 included 7 regulators more abundant in Pi limitation than in Pi proficiency at 48 h and 60 h in *SL* and *SC*, respectively, and more abundant in *SL* than in *SC* in Pi proficiency. This cluster includes the SHK OsaA/SCO5748 of the TCS OsaA-OsaB (cluster A2) involved in osmoadaptation and induced after OsS in a sigma B-dependent manner [66]. It is noteworthy that OsaA (cluster B3) and OsaB (cluster A2) do not constitute a classical TCS since they are not co-transcribed, and they show a different abundance pattern in the two strains. SCO3209, which is involved in the regulation of para-hydroxybenzoate catabolism [93], also belongs to this sub-cluster.

#### 2.2.3. Cluster C: Regulators More Abundant in *SC* than in *SL* in Both Pi Conditions

Cluster C includes 53 regulators (Figure 4):-Two regulatory systems able to sense nitric oxide (NO), the TCS OdsK/SCO0203-OdsR/SCO0204 [94] also called DevS/R [95], and NsrR/SCO7427 which positively controls the expression of NO-detoxifying flavohemoglobins hmpA1/SCO7428 and hpmA2/SCO7094 [96,97]. The RR OdsR regulates the expression of genes of the dormancy/survival regulon [94]. Proteins of the OdsR regulon fall into two groups, highly and poorly expressed in *SC*, and conversely in *SL* (Figure S11C of [9]). Intracellular NO is thought to inhibit the auto-phosphorylation of the SHK DevS/OdsK which becomes unable to phosphorylate the RR DevR/OdsR [95]. Un-phosphorylated OdsR is unable to interact with the promoter region of *actII-ORF4* to activate its expression [95]. NO thus has an indirect negative effect on ACT biosynthesis. Since *SC* produces ACT abundantly, it might be legitimate to propose that little NO is generated in *SC*, and this would be consistent with the previously mentioned reduced N availability in this strain.

Cluster C also includes 2two regulators involved in the response to OsS or OxS:-The SHK OhkA/SCO1596 and its cognate RR OrrA/SCO3008 (Cluster D2). This TCS is involved in osmoadaptation [98] and has a negative impact on RED and ACT biosynthesis [99,100].-The regulator WblA that is a target of the OrrA was identified as a down-regulator of the expression of genes involved in the resistance to OxS [101]. Since OxS was proposed to be an important trigger of ACT biosynthesis [8,12], this can explain the negative impact that WblA exerts on ACT biosynthesis [102].-The protein kinase RsfA/SCO4677 that phosphorylates and thus inactivates the function of the anti-sigma factor antagonist BldG/SCO3549 [25] that is involved in the regulation of the activity of the OsS responsive SigH factor [26].

**Figure 4 ijms-23-14792-f004:**
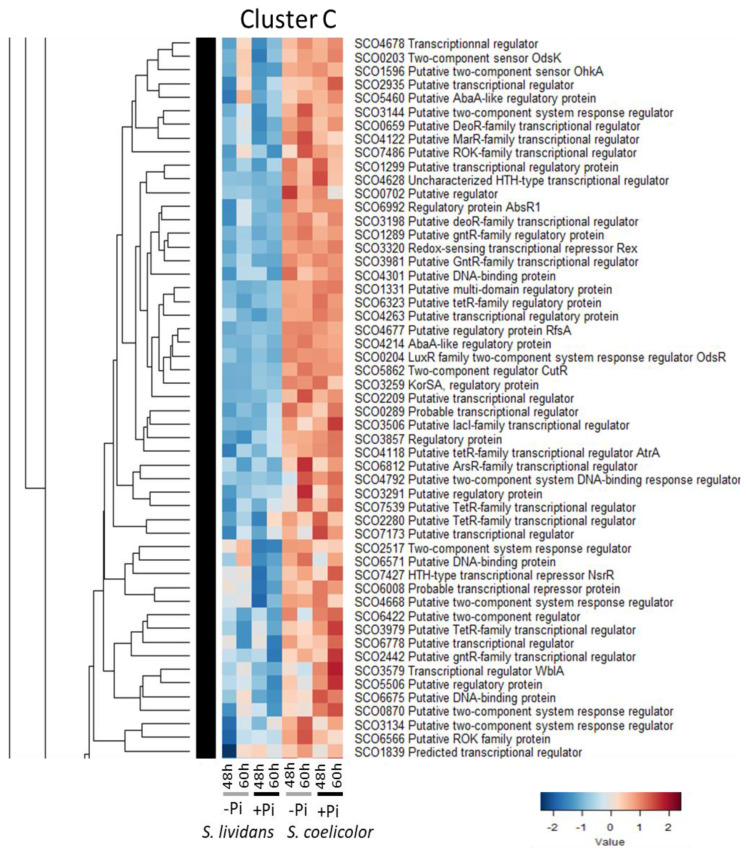
Cluster C, heatmap representation of transcriptional regulators and sensory histidine kinases more abundant in *SC* than in *SL* in both Pi conditions.


-The regulator Rex SCO3320 [103] that responds to NAD+/NADH poise and represses the transcription of numerous genes, including its own and that of genes encoding subunits of the complex I/NADH dehydrogenase of the respiratory chain, (*nuoABCDEFGHIJKLMN*/*sco4562-75*) and of the ATP synthase (*atpIBEFHAGDC*/*sco5366-74*). The repressing effect of Rex is alleviated when the intracellular NADH concentration is high [103], as is likely to be the case in *SC* that is characterized by an oxidative metabolism [10]. Interestingly, the sub-units of the complex I of the respiratory chain were far more abundant in *SC* than in *SL*, in both Pi conditions (Figure 5 of [9]).-SCO6992/AbsR1, a member of the AbrC3 regulon [76,78] that has a positive impact on RED and ACT biosynthesis [104].-The RR CutR/SCO5862 of the TCS CutS/CutR, thought to be involved in the regulation of copper homeostasis [105], has a negative impact on RED and ACT biosynthesis [106].-SCO6008/Rok7B7 that negatively regulates the expression of the xylose transporter operon *xylFGH*, has a rather a positive impact on ACT biosynthesis but a negative one on RED and CDA biosynthesis [107,108].


#### 2.2.4. Cluster D: Regulatory Proteins More Abundant in Pi Proficiency than in Pi Limitation in *SC*

Cluster D could be divided into three sub-clusters: D1 and D2 and D3 (Figure 5). Most of these regulators have a positive or a negative impact on antibiotic production.

The 18 regulators of sub-cluster D1 were more abundant in Pi proficiency than in Pi limitation at both time points in *SC* but some of them were also abundant in Pi limitation at 48 h in *SC*. This sub-cluster includes:-The regulator DasR/SCO5231 which is known to repress the expression of genes involved in the up-take and catabolism of N-acetyl glucosamine [109]. Glucosamine-6-phosphate (GlcN-6P) and N-acetylglucosamine-6-phosphate (GlcNAc-6P) act as allosteric effectors of DasR impairing binding to operator sites and thus allowing the expression of the DasR-target genes. Since several enzymes involved in cell wall degradation were far more abundant in *SC* than in *SL* (Figure S10, cluster C of [9]), GlcN-6P as GlcNAc-6P are likely to result from the autolytic degradation of peptidoglycan that obviously takes place in *SC* but not in *SL*.-The RR MacR/SCO2120 of the TCS MarS/MarR positively controls the expression of several membrane proteins involved in maintenance of cell wall integrity of some proteins involved in primary carbon metabolism and of proteins of the Zur regulon which plays a role in zinc homeostasis [110,111]. Furthermore, MacR was shown to directly interact with the *actII-orf4* promotor region and to have a positive impact on ACT, as well as CDA and RED biosynthesis [111].-The RR RapA1/SCO5403 of the TCS RapA2/RapA1, which has a positive impact on CPK and ACT biosynthesis [112].-BldC/SCO4091, which has a positive impact on antibiotic production and morphological differentiation [113,114].-The ScbR-like regulator SlbR/SCO0608 which, in interaction with γ-butyrolactone SCB1, binds to the promoter regions of *scbR/A* and *adpA* [115] and inhibits the transcription of these genes that positively control morphological development and antibiotic production [116]. SlbR thus has an indirect negative impact on antibiotic production and morphological differentiation [117].

The 18 regulators of sub-cluster D2 were more abundant in Pi proficiency than in Pi limitation at both time points in *SC* but some of them were also abundant in Pi limitation at 60 h in *SL*. This sub-cluster includes:-The RR OrrA/SCO3008 of the TCS OhkA(SCO1596, cluster C)/OrrA involved in osmoadaptation that has a negative impact on RED and ACT biosynthesis [99,100].-The starvation sensing protein SpaA/SCO7629, an orthologue of the *E. coli* protein RpsA involved in an homoserine lactone signaling pathway [118] that has a positive impact on antibiotic biosynthesis [119].-The gene encoding the regulator SCO2152 is located in divergence of the *qcrCAB* operon (*sco2148—sco2151*), which plays an important role in growth and development of *SC* in conditions of oxygen limitation [120,121]. SCO2152 was rather more abundant in *SC* than in *SL* in both Pi conditions, whereas QcrA/SCO2149 showed an opposite trend (Figure 5 of [9] and Appendix A) suggesting that SCO2152 might negatively regulate the expression of the *qcrCAB* operon.

**Figure 5 ijms-23-14792-f005:**
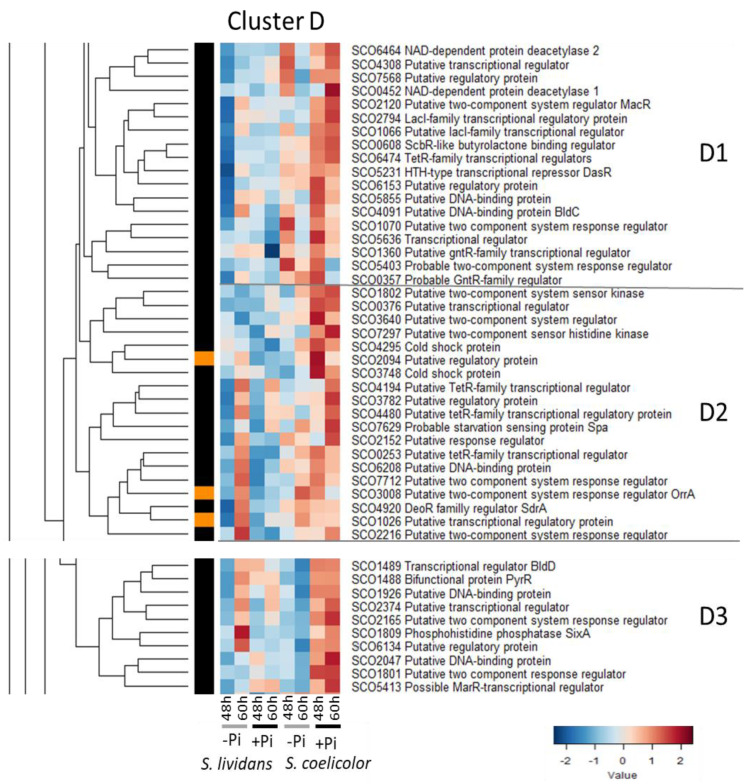
Cluster D, heatmap representation of transcriptional regulators and sensory histidine kinases more abundant in Pi proficiency than in Pi limitation in *SC*. Cluster D that can be divided into 3 sub-clusters D1, D2 and D3. The 18 regulators of sub-cluster D1 were more abundant in Pi proficiency than in Pi limitation at both time points in *SC* but some of them were also abundant in Pi limitation at 48 h in *SC*. The 18 regulators of sub-cluster D2 were more abundant in Pi proficiency than in Pi limitation at both time points in *SC* but some of them were also abundant in Pi limitation at 60 h in *SL*. The 10 regulators of sub-cluster D3 were more abundant in Pi proficiency than in Pi limitation at both time points in *SC* but some of them were also abundant in Pi limitation at 60 h in *SL*.

The 10 regulators of sub-cluster D3 were also more abundant in Pi proficiency than in Pi limitation at both time points in *SC* but some of them were also abundant in Pi limitation at 60 h in *SL*. This cluster includes the phosphohistidine phosphatase SixA/SCO1809 whose function is unknown in Streptomyces and the pleiotropic regulator BldD/SCO1489, whose DNA binding activity is modulated by the signaling molecule c-di-GMP [122,123]. BldD has a positive impact on antibiotic production and morphological differentiation [124]. The phenotype of the bldD mutant is thought to be partly due to an increase in expression of the pleiotropic regulatory gene wblA (cluster C) [125], a down-regulator of the expression of genes involved in the resistance to OxS [101].

#### 2.2.5. Cluster E: Regulatory Proteins More Abundant in Pi Limitation than in Pi proficiency in *SC*

Cluster E can be divided in two sub-clusters: E1 and E2 (Figure 6).

The 23 regulators of sub-cluster E1 were clearly more abundant in Pi limitation than in Pi proficiency in *SC* but not in *SL*. Sub-cluster E1 includes:-SCO7327/OsaD, thought to be involved in OsS sensing [66,126].-The zinc-binding regulator HypR/SCO6294 [127], whose expression is induced by hydroxyproline. SCO6294 controls the expression of the Hyp operon (*sco6289-sco6293*) involved with L-hydroxyproline catabolism. Since hydroxyproline is known to enhance the activity of the peptidoglycan N-deacetylase [128], the induction of HypR expression in *SC* in Pi limitation might be related to cell wall alteration in *SC*.

**Figure 6 ijms-23-14792-f006:**
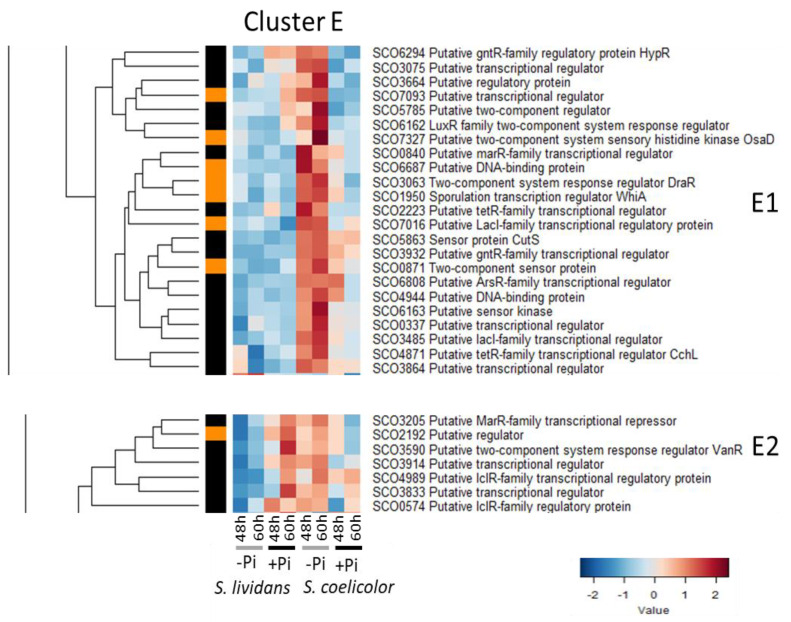
Cluster E, heatmap representation of transcriptional regulators and sensory histidine kinases more abundant in Pi limitation than in Pi proficiency in *SC*. Cluster E can be divided into 2 sub-clusters, E1 and E2. The 23 regulators of sub-cluster E1 were more abundant in Pi limitation than in Pi proficiency in *SC* but not in *SL*. The 7 regulators of sub-cluster E2 were more abundant in Pi limitation than in Pi proficiency in *SC* but were also abundant at 60 h in Pi proficiency in *SL*.

This cluster also includes several regulators playing in role in the regulation of antibiotic biosynthesis and/or morphological differentiation, such as:-The RR DraR/SCO3063 of the TCS DraK (sub-cluster A2)/DraR, which has a positive impact on ACT biosynthesis but a negative one on CPK and RED biosynthesis [62].-The ArsR-like regulator SCO6808, which has a negative impact on antibiotic biosynthesis [117]-The sensory HK CutS/SCO5863 of the TCS CutR (cluster C1)/CutS, which is thought to be involved in the regulation of copper homeostasis [105] and has a negative impact on RED and ACT biosynthesis [106].-The KorSA-like regulator SCO3932 is located upstream of a pSAM2-like ICE element and has a positive impact CPK and ACT synthesis via its direct interaction with *cpkD*, a coelimycin biosynthetic gene, and *actII-orf4* [129].-The RR SCO5785 of the TCS SCO5784/SCO5785 has a positive impact on the production of several extracellular proteins and on antibiotics biosynthesis, and a negative one on the expression of some ribosomal genes [130].-The MmyB-like regulator SCO4944 is a putative member of the A-factor signaling cascade [131] and is likely to have a positive impact on antibiotic biosynthesis.-The protein WhiA/SCO1950 which governs spore formation [132,133].

The 7 regulators of sub-cluster E2 were also more abundant in Pi limitation than in Pi proficiency in *SC* but were also abundant at 60 h in Pi proficiency in *SL*. This cluster includes *sco3205* encoding a regulator of unknown function [134,135], as well as *vanR*/*sco3590*, encoding an activator of the expression of the divergently located *vanJKHAX* cluster conferring inducible vancomycin resistance [136,137]. The low abundance of SCO3590 in Pi proficiency in *SC* could be seen as consistent with the reported hyper-sensitivity of *SC* to vancomycin in Pi proficiency [138]. The VanRS and CseBC TCS (sub-cluster B1) are both involved in the signaling of peptidoglycan/cell wall alteration [136,137,139].

#### 2.2.6. Cluster F: Regulators More Abundant in *SL* than in *SC* in Both Pi Conditions

Cluster F includes 44 regulatory proteins more abundant in *SL* than in *SC* in both Pi conditions (Figure 7). This cluster includes regulators known to have either a positive or a negative impact on antibiotic biosynthesis and/or morphological biosynthesis in *SC* and some of them are involved in the regulation of metal homeostasis.

Regulators with a positive impact on antibiotic biosynthesis and/or morphological differentiation include:-Zur/SCO2508 which, in interaction with Zinc [140], positively controls the expression of the gene encoding the zinc exporter ZitB/SCO6751 [141] and negatively that of the genes encoding an ABC transporter involved in Zn uptake (SCO2505/SCO2506/SCO2507) [141,142], and of a cluster of genes directing the biosynthesis of the zincophore, coelibactin [143]. Consistently, ZitB/SCO6751 was poorly abundant in *SC* and in *SL*, and to a lesser extent in Pi proficiency (Figure S19 of [9]), whereas proteins of the coelibactin cluster and SCO2506 were more abundant in Pi proficiency than in Pi limitation in *SC* as in *SL*, to a lesser extent (Figure 10 of [9]). This indicated that Zur is involved in the regulation of Zn homeostasis, maintaining a low intracellular Zn concentration. In *S. avermitilis*, the deletion of *zur* resulted in decreased production of antibiotics and delayed morphological differentiation [144], as did the addition of Zn in the growth medium of *SC* [145] indicating that Zur plays a positive role in the regulation of antibiotic biosynthesis.

**Figure 7 ijms-23-14792-f007:**
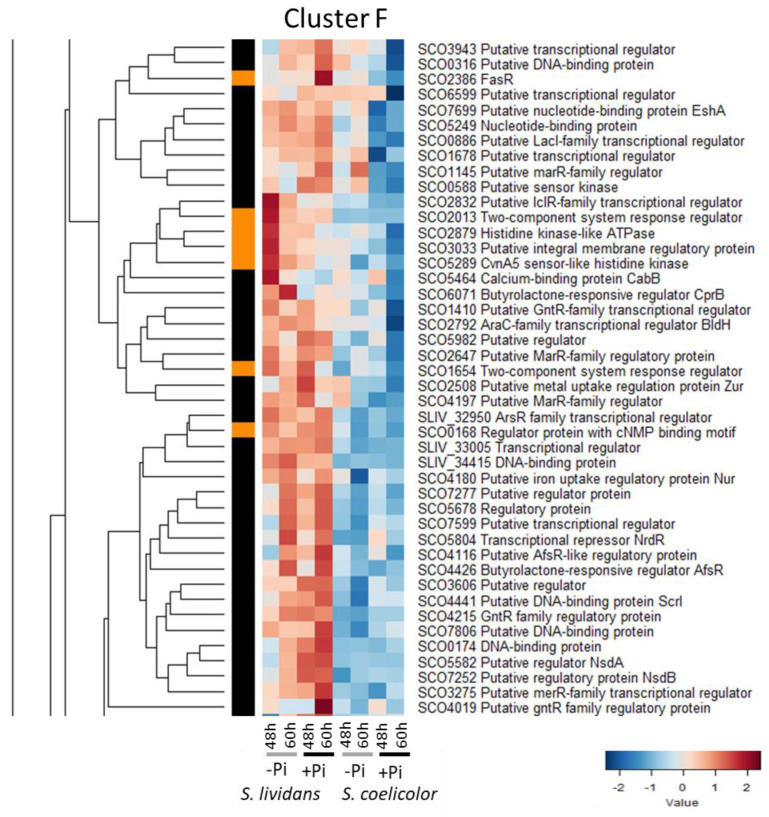
Cluster F, heatmap representation of transcriptional regulators and sensory histidine kinases more abundant in SL than in SC in both Pi conditions.


-The nickel-responsive regulator SCO4180/Nur that controls the expression of nickel transport and of Fe or Ni-dependent superoxide dismutases [146,147,148].-SCO5464/CabB, a calmodulin-like protein that has a role in calcium homeostasis [149].-The A-factor dependent regulator AfsR/SCO4426 [150,151,152], a global regulator, that has a positive impact on antibiotic biosynthesis [153] and morphological differentiation [154,155].-Scr1/SCO4441, which has a strong positive impact on endogenous antibiotic production of *SC*, *SL*, and other *Streptomyces* strains [156].-The nucleotide binding protein SCO7699/EshA, which stimulates the accumulation of ppGpp [157,158].-The regulator SCO1678, which negatively controls the expression of the divergent operon involved in gluconate metabolism [159].


Regulators with a negative impact on antibiotic biosynthesis and/or morphological differentiation include:-The A-factor receptor homolog CprB/SCO6071, which has a negative impact on γ-butyrolactone biosynthesis and thus indirectly on ACT biosynthesis and morphological differentiation [160,161].-BldH/SCO2792/AdpA, which activates the transcription of *ramR*, an atypical response regulator that itself activates expression of the genes of the *ramCSAB* operon required for aerial mycelium formation [162].-NsdB/SCO7252 [163] and NsdA/SCO5582 [164,165]. NsdA was more abundant in Pi proficiency than in Pi limitation in *SL*. *nsdA* is located just upstream of a cluster of genes targets of GlnR/SCO4159, an important regulator of N metabolism [166]: *amtB*/*sco5583* (ammonium transporter), *glnK*/*sco5584* (regulatory protein P-II, cluster I), and *glnD*/*sco5585* (protein PII adenylyltransferase) [167]. These proteins were more abundant in Pi proficiency than in Pi limitation in *SL* as well as in *SC* but to a far lesser extent (Figure 7 of [9]). Considering the proximity of *ndsA* with this cluster of genes and its differential abundance in *SL* and *SC*, NdsA might play a role in the regulation of the expression of genes of this cluster either directly or indirectly via the regulation of *glnR* expression.

Cluster F also includes:-The positive regulator of fatty acid biosynthesis, FasR/SCO2366 [168]. FasR was far less abundant in *SC* than in *SL*, in Pi proficiency, consistent with the previously reported lower phospholipid content of *SC* compared to *SL* [169].-SCO5804/NrdR, which represses the expression of class I and class II ribonucleotide reductases encoding genes (RNRs, *nrdAB*/*sco5225-26*, and *nrdJ*/*sco5805*) involved in the conversion of ribonucleotides to deoxyribonucleotides [170,171]. NrdR was similarly abundant in both Pi conditions in *SL* whereas NrdAB were clearly more abundant in Pi proficiency than in Pi limitation in *SL* (Figure S7A of [9]) but far less abundant in *SC* than in *SL* in both Pi conditions (Figure S7A of [9]) suggesting a default in the activation of the expression of these genes in *SC*.-SCO2647 bears similarities with PecS from plant pathogens [172] and is responsive to urate that is generated by xanthine dehydrogenase at the same time as reactive oxygen species (ROS). SCO2647 might thus be induced in conditions of OxS.-The gene encoding SCO2832 is located in divergence of an operon encoding a probable amino acid ABC transporter (*sco2828* to *sco2831*). SCO2828 (Figure S18B of [9]) and SCO2830 (Figure S15D of [9]) were abundant in *SC* but not in *SL* in Pi limitation suggesting that SCO2832 represses their expression.

#### 2.2.7. Cluster G: Regulatory Proteins More Abundant in Pi Proficiency than in Pi Limitation in *SL* but Not in *SC*

This cluster includes 14 regulatory proteins (Figure 8):-The nitrogen regulator GlnRI/SCO4159, which activates the expression of genes involved in nitrogen/uptake and assimilation [70,86,173] but directly represses the expression of *actII-ORF4*, whereas it activates that of *redZ*, activator of the ACT and RED pathways, respectively [174]).-The gene encoding the TetR regulator SCO2775 is located upstream of genes (*sco2774-73*) encoding proteins involved in the β-oxidation of fatty acids and in divergence of genes encoding acetyl/propionyl CoA carboxylase (α/SCO2777 and β/SCO2776 subunits) involved in the biosynthesis of malonyl or methyl-malonylCoA, possibly used for fatty acids and/or polyketide (ACT) biosynthesis. SCO2777 and SCO2776 were highly abundant in *SC* but poorly abundant in *SL*, in both Pi conditions (Figure S2A of [9]), suggesting that SCO2775 represses the expression of the corresponding encoding genes, consistently with high ACT production of *SC*.-The orphan atypical RR SCO4768/BldM [175] positively controls the formation of aerial mycelium [176].

**Figure 8 ijms-23-14792-f008:**
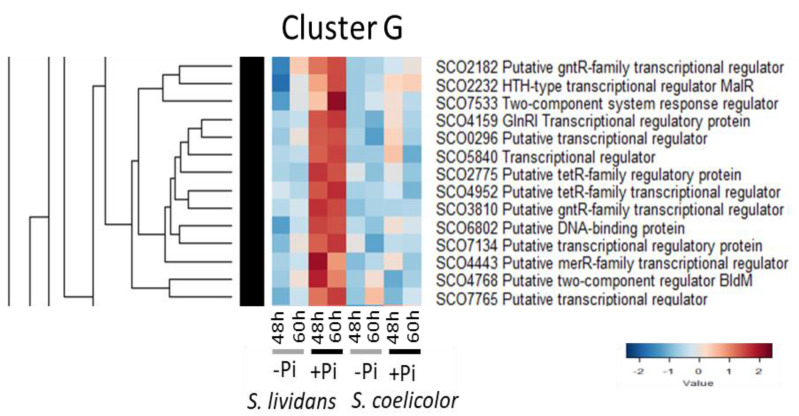
Cluster G, heatmap representation of transcriptional regulators and sensory histidine kinases more abundant in Pi proficiency than in Pi limitation in *SL* but not in *SC*.


-The gene encoding the regulator SCO4443 is located downstream of a gene encoding a glutathione peroxidase/SCO4444. SCO4444 was more abundant in Pi limitation than in Pi proficiency in *SL* (Figure S11A of [9]) (but it was the opposite in *SC*) suggesting that SCO4443 represses SCO4444 expression in Pi proficiency, at least in *SL*.-The gene encoding the regulator MalR/SCO2232 is located in divergence of the maltose transport/utilization operon [177,178] and mediates maltose induction and glucose repression of the maltose transport/utilization operon [179]. Consistently, SCO2231/MalE, putative maltose-binding protein, was mainly abundant in Pi limitation in *SL* (Figure S15B of [9]). This regulation is obviously altered in *SC* since MalE was abundant in both Pi conditions in *SC*. In contrast, the maltose permease SCO2229/MalG (Figure S15D of [9]) was mainly abundant in Pi limitation in *SC*.-The gene encoding the regulator SCO2182 is located upstream of *sco2183*/*aceE1* encoding a putative component E1 of pyruvate dehydrogenase. SCO2183 was similarly abundant in the two strains in Pi proficiency (Figure S1 of [9]) suggesting that SCO2182 positively regulates its expression in Pi proficiency.


#### 2.2.8. Cluster H: Regulatory Proteins More Abundant in Pi Limitation than in Pi Proficiency in *SL* but Not in *SC*

This cluster includes 11 regulatory proteins (Figure 9):-The response regulator MtrA/SCO3013 of the TCS SCO3013/SCO3012 (MtrB, HK) which, as PhoP, regulates both N and Pi metabolism. MtrA negatively auto-regulates its own expression and that of GlnR/SCO4159 [180,181]. MtrA represses, in condition of N proficiency, the genes that are activated by GlnR/SCO4159 in condition of N limitation [181]. MtrA has a direct, but either positive or negative, impact on the expression of PhoRP and of other genes of Pi metabolism depending on the composition of the growth medium [182]. MtrA also binds to sites located in the promoter regions of genes encoding pathway specific activators ActII-Orf-1, ActII-Orf4, and RedZ and has a negative impact on their expression and thus on ACT and RED biosynthesis [183]. It also represses the expression of *bldD* (sub-cluster D3) that plays a positive role in the regulation of antibiotic biosynthesis [125]. In contrast, MtrA activates the expression of genes involved in formation of aerial mycelium, including *chp*, *rdl*, and *ram* and regulatory genes of the Bld and Whi families [184]. MtrA thus positively controls morphological differentiation and negatively antibiotic production.

**Figure 9 ijms-23-14792-f009:**
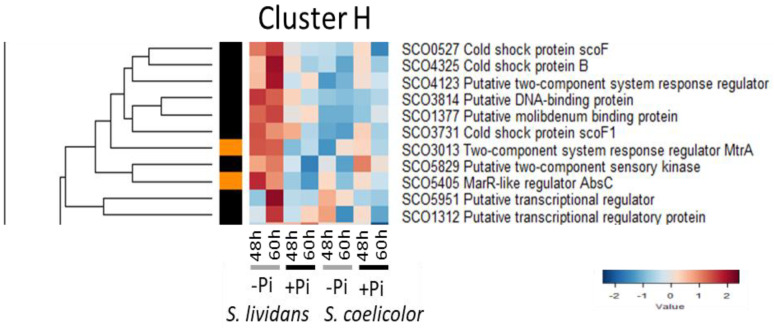
Cluster H, heatmap representation of transcriptional regulators and sensory histidine kinases more abundant in Pi limitation than in Pi proficiency in *SL* but not in *SC*.


-The regulator AbsC/SCO5405 plays a role in Zinc (Zn) homeostasis since it negatively controls, together with Zur (cluster F), the expression of the cluster directing the biosynthesis of the zincophore, coelibactin, and of another NRPS cluster with predicted siderophore-like activity [185]. The deletion of AbsC results in elevated expression of these clusters and thus higher Zn availability. Since Zn was shown to have a negative impact on antibiotic biosynthesis [145], the negative impact that AbsC exerts on antibiotic production is thought to be due to enhanced Zn availability [185]. Furthermore, numerous enzymes belonging to carbon and nitrogen metabolism were more abundant in an AbsC mutant suggesting that AbsC negatively regulates their expression [186]. Consistently, most of the enzymes of central carbon and nitrogen metabolism listed in [186] and thought to be under the negative control of AbsC were indeed more abundant in Pi proficiency, when the expression of AbsC is low [9], than in Pi limitation. The cause of the low abundance of AbsC in *SC* is unknown but is likely to have important consequences on the cellular metabolism of this strain.-The gene encoding the sensory HK SCO5829 of the TCS SCO5829/SC5828 (RR) is located downstream of a gene encoding sucrase/ferredoxin-like protein SCO5830 and SCO5831 a citrate synthase-like protein. SCO5831 was clearly more abundant in Pi proficiency than in Pi limitation in *SL* (Figure 4 of [9]) suggesting that this TCS might repress SCO5831 expression in Pi limitation in *SL*.-Finally, the three cold shock proteins (Csps) SCO0527/ScoF, SCO3731/ScoF1, and SCO4325/CspB were also highly expressed in *SL* in Pi limitation. Originally, Csps were viewed as nucleic acid chaperons preventing the formation of secondary structures in mRNA at low temperature and thus facilitating the initiation of translation but since the expression of some Csps was shown to be non-cold inducible, they are now viewed as involved in the adaptation to various stresses [187].


### 2.3. Eukaryotic-Like Serine or Threonine Protein Kinases

The genome of *S. coelicolor* and *S. lividans* contain numerous genes encoding eukaryotic-like serine or threonine protein kinases (ESTPK) [188,189]. In this study, 23 ESTPK were detected and can be classified into three main clusters (a, b, c, Figure 10).

The 5 proteins of cluster a were more abundant in *SL* than in *SC* in Pi proficiency or limitation. Two proteins were rather more abundant in Pi proficiency than in Pi limitation, whereas it was the opposite for 3 others, including the putative membrane-associated sensory kinase RamC/SCO6681 [190]. RamC is encoded by the first gene of the *ramCSAB* operon that includes *ramS*/*sapB* encoding the morphogenetic peptide RamS and components of an ABC transporter (RamAB). These proteins are necessary for morphological differentiation but not for vegetative growth nor antibiotic biosynthesis and their expression is under the positive control of the response regulator RamR [162,191].

**Figure 10 ijms-23-14792-f010:**
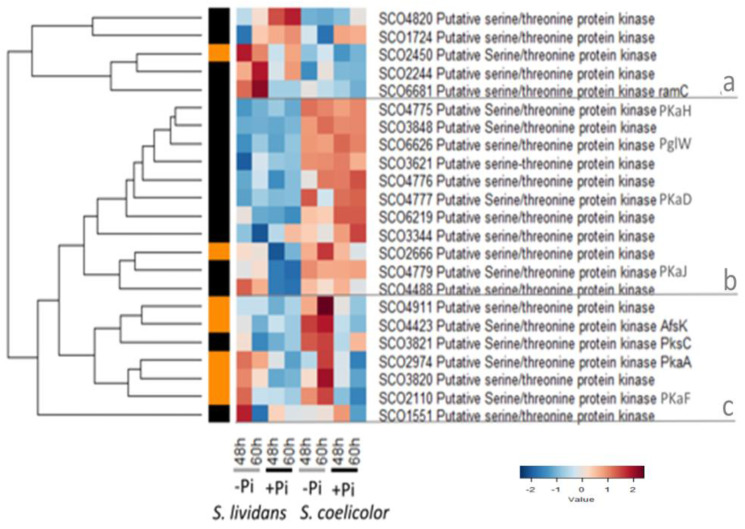
Heatmap representation of the abundance of eukaryotic-like serine or threonine protein kinases in *SL* and *SC* that can be divided in 3 sub-clusters, a, b and c. The 5 proteins of cluster a were more abundant in *SL* than in *SC* in Pi proficiency or limitation. The 11 proteins of cluster b were more abundant in *SC* than in *SL* in both Pi conditions. The 7 proteins of cluster c were more abundant in *SC* than in *SL* in Pi limitation at both time points but 4 of these proteins were also abundant in Pi limitation, in *SL* at 48 h.

The 11 proteins of cluster b were more abundant in *SC* than in *SL* in both Pi conditions. This cluster includes four of the five kinases belonging to the sco4775-sco4779 cluster encoding PkaH/SCO4775, SCO4776, PkaD/SCO4777, PkaI/SCO4778 (not detected), and PkaJ/SCO4779. Two of these EPK (PkaH and PkaI) were shown to be part of the spore wall synthesizing multi-protein complex SSSC which includes proteins directing peptidoglycan (MreBCD, PBP2, Sfr, RodZ) and cell wall glycopolymer synthesis (PdtA) [192]. PkaI and PkaH/SCO4775 were shown to phosphorylate MreC [193]. The high abundance of these proteins in *SC* might be related to the cell wall stress occurring in these bacteria. This cluster also includes PglW/SCO6626, which, together with PglY (ATPase), PglX (DNA methylase), and PglZ (inhibitor of PglX activity), is part of a phage growth limitation (Pgl) system that confers protection against the temperate bacteriophage phiC31 and its relatives [194]. PglW was proposed to transduce a signal, probably via phosphorylation, to other Pgl proteins resulting in the activation of the DNA methyltransferase PglX leading to phage restriction [195].

The 7 proteins of cluster c were more abundant in *SC* than in *SL* but mainly in Pi limitation at both time points. However, 4 of these proteins were also abundant in Pi limitation, in *SL* at 48 h. This cluster includes:-AfsK/SCO4423 which phosphorylates, together with other ESTPK [196], AfsR/SCO4426, a global regulator, that has a positive impact on antibiotic biosynthesis [153] and morphological differentiation [154,155]. AfsK was also shown to phosphorylate the DnaA initiator protein impairing its interaction with the replication origin and thus inhibiting replication [197].-PkaF/SCO2110, whose over-expression was shown to be correlated with a strong reduction of ACT (but not of RED) biosynthesis and an inhibition of morphological differentiation, suggesting that it plays a negative role in the regulation of these processes [198].

## 3. Discussion

In this study, the abundance of 332 proteins, including proteins of the transcriptional apparatus and one or two components transcriptional regulators, was quantified in *SL* and *SC* grown in conditions of either phosphate limitation or proficiency. The abundance of the vast majority of these proteins greatly differed between the two strains and responded differently to Pi availability, whereas only a minority had a similar abundance and responded in a similar way to Pi availability in both strains.

An important outcome of this paper is that the regulatory proteins GlnRI/SCO4159/SLI_4154 (cluster G) [70,86,173] and MtrA/SCO3013/SLI_3005 (cluster H) [180,181], involved in the positive and negative control of nitrogen metabolism, respectively, were poorly abundant in *SC* compared to *SL* and this was correlated with the very low abundance of proteins involved in nitrogen up-take and assimilation in *SC* compared to *SL*, as reported in Lejeune et al. [9]. Consequently, *SC* most likely suffers from severe nitrogen stress. GlnRI activates the expression of genes involved in N uptake and assimilation [86] and is, as expected, more abundant in Pi proficiency than in Pi limitation in *SL*, since in this condition the negative effect that PhoP exerts on GlnRI expression is relieved [199]. However, this up-regulation does not occur in *SC*. Consistently, proteins whose expression is under the positive control of GlnR such as the ammonium transporter AmtB/SCO5583, the glutamine synthases GSI/SCO2198/GlnA and GSII/SCO2210, as well as the regulator SCO2958/NnaR and its regulatory targets essential for nitrate and nitrite assimilation [70,86,173] were far more abundant in Pi proficiency than in Pi limitation in *SL* but not in *SC* (Figure 8 of [9]). In contrast, the regulator MtrA, which represses the expression of glnRI and that of the genes under its positive control [181], is clearly more abundant in Pi limitation than in Pi proficiency in *SL* but not in *SC*. This study thus revealed that GlnRI that positively controls the expression of genes involved in N up-take and assimilation is more abundant in Pi proficiency than in Pi limitation, whereas MtrA, which negatively controls the expression of these genes, is more abundant in Pi limitation than in Pi proficiency, in *SL*. The interplay between these two regulators thus adjusts N up-take and assimilation to Pi availability. However, in *SC*, such regulation is altered, and these two regulators remain poorly abundant in both Pi conditions. This is obviously correlated with the poor expression of genes encoding proteins involved in nitrogen up-take and assimilation. Interestingly, these two regulators are known to be phosphorylated and this phosphorylation stimulates the repressing function of MtrA [200] but impairs the activating function of GlnRI [201]. In *SC*, the phosphorylation of MtrA could thus contribute to the weak expression of GlnRI and the activating function of the latter would be impaired by its phosphorylation, resulting in poor expression of its target genes (Figure 7 of [9]). We hypothesize that the extensive phosphorylation of these regulators might be promoted in *SC* by the high ATP content of this strain [10], which is thought to be linked to the highly active oxidative metabolism of this strain [9,12], as well as to the high abundance of the complex I of the respiratory chain in *SC*, in both Pi conditions (Figure 5 of [9]).

The reduced abundance, in *SC* compared to *SL*, of the glutamine synthases GlnA/SCO2198 and GSII/SCO2210, to a lesser extent (Figures 6 and S3A of [9]), and of other enzymes involved directly or indirectly (via serine biosynthesis) in the biosynthesis of alanine and glycine (Figure S3A of [9]) is likely to limit the availability of these three amino acids as well as that of the N containing compound, N acetyl glucosamine (GlcNAc), which are constitutive of the peptidoglycan cell wall [202]. The high abundance in *SC*, in both Pi conditions, of several enzymes involved in cell wall degradation belonging to the DasR regulon (Figure S10, cluster C of [9]) indicated an autolytic degradation of the peptidoglycan cell wall in *SC*. This degradation would result in the generation of glucosamine-6P and N-acetylglucosamine-6P, the allosteric effectors of DasR [203], which impair the interaction of the latter with its operator sites allowing the expression of genes involved in N-acetylglucosamine (NAG) metabolism which is under the negative control of DasR [204]. We propose that in *SC* the degradation of the peptidoglycan cell wall would be triggered by nitrogen stress in order to recycle the N present in the constituents of the peptidoglycan cell wall, and would be sensed as “cell wall stress” inducing the expression of the SigE and SigU sigma factors (Figure 1) as well as that of the TCS vanR/vanS (sub-cluster E2) and cseB/cseC (sub-cluster B1) (Table 1).

A structurally altered peptidoglycan cell wall should not be able to fulfill its osmo-protective function and this would result in osmotic stress (OsS) triggering the expression of SigH, the anti SigH SCO5244 and anti sigma factor antagonist BldG, as well as the TCS OhkA/OrrA involved in OsS response (Table 1). Consistently, electron microscopic pictures of *SL* and *SC* show that mycelial fragments of SC were swollen compared to those of *SL*, testifying to water entry (Figure 3B of [205]).

Our study thus indicates that nitrogen, cell wall, and osmotic stresses are more intense in *SC* than in *SL* and are intimately linked, with nitrogen stress being the most likely initial cause of cell wall and osmotic stresses. Interestingly, the higher abundance of regulators positively controlling the expression of genes involved in the resistance to cell wall and osmotic stresses in *SC* compared to *SL* (14 versus 4, Table 1), signals an attempt of *SC* to combat these stresses to maintain its cellular homeostasis. Most of these regulators have a negative impact on antibiotic biosynthesis so when these regulators are inactivated, these stresses are getting worse, and this obviously leads to a further enhancement of antibiotic production (Table 1). This observation raises the question of the nature of the links existing between these different stresses and antibiotic production. To answer this question, we propose that when the bacteria is facing a stressful situation, its attempts to combat these stresses to maintain its homeostasis are ATP consuming. The permanence and intensity of these stresses would possibly overwhelm these resistance systems leading to further ATP consumption to face the detrimental consequences of the latter, resulting in ATP depletion. Such ATP depletion would trigger the activation of the oxidative metabolism that, coupled with respiration, constitutes an attempt to restore the cellular energetic balance but also generates OxS [9,10,12]. OxS was proposed to be an important trigger for the expression of ACT in *SC* [9,12] since ACT was shown to bear an anti-oxidant function [8]. The anti-oxidant function of ACT is thought to be due to its ability to capture excess electrons of ROS and NOS thanks to its quinone groups [206]. Furthermore, since the onset of ACT biosynthesis was shown to coincide with an abrupt drop of the ATP content of the bacteria [10], ACT might also be able to capture electrons of the respiratory chain, reducing its activity and thus the generation of ATP. ACT would thus have both anti-respiratory and anti-oxidant functions. The two other antibiotics produced by *SC*, CDA [207], and RED [208] are known to alter the permeability of the cellular membrane and this might lead to the dissipation of the H+ gradient and thus to a reduction of respiratory activity and ATP generation. The function of these three molecules would be to reduce the excessive respiratory activity of *SC* in order to limit its detrimental effects such as OxS as well as the generation of ATP when Pi is scarce.

Interestingly, 49 regulators among 71 (69%) were more abundant in *SC* than in *SL* but approximately the same number were reported to have a positive (17) or a negative (14) impact on antibiotic biosynthesis. These regulators are either known to directly interact with the promoter region of specific regulators or biosynthetic genes of the antibiotic biosynthetic pathways (Class I, in bold text in Table 1) or their contribution to the regulation of antibiotic biosynthesis is indirect and often un-elucidated (Class II, in plain text in Table 1). This indicated that antagonistic processes are concomitantly at work, in *SC*, to optimize the ad hoc level of antibiotic production to fulfill their proposed role in the regulation of the energetic metabolism of the bacteria [12]. Our conception of the relationships between the physiological and metabolic features of *SC* and the production of antibiotics is summarized in the graphical representation of Figure 11. In this model, antibiotic production is proposed to be triggered by high OxS resulting from highly active respiratory activity linked to the activation of the oxidative metabolism, to the high abundance of enzymes of the complex 1 of the respiratory chain [9], as well as to the reduced availability of NADPH, a co-factor playing a key role in the resistance to OxS. The low NAPDH availability was proposed to be due to the previously reported low carbon flux through the pentose phosphate pathway in *SC* [9], whereas the activation of the oxidative metabolism, aimed at ATP replenishment, would be triggered by the ATP depletion resulting from various stresses and phosphate limitation.

**Figure 11 ijms-23-14792-f011:**
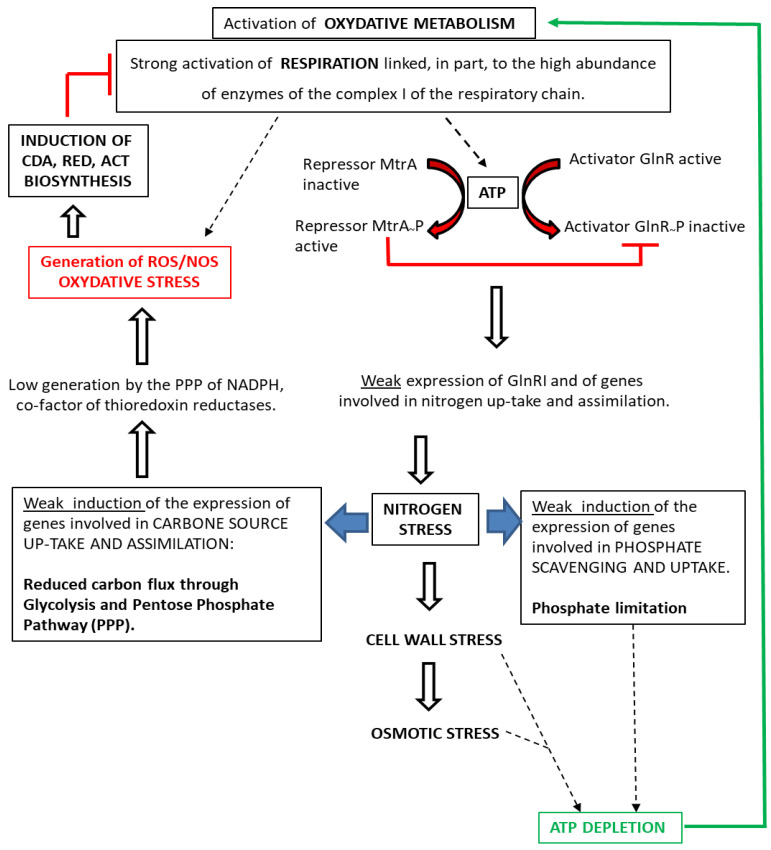
Schematic representation of the proposed systemic understanding of the relationships between specific physiological and metabolic features of *SC* and antibiotics production.

**Table 1 ijms-23-14792-t001:** List of the 71 transcriptional regulators and 23 eukaryotic-like protein kinases detected in our study and classified, when possible, according to their role in the regulation of phosphate and/or nitrogen up-take and assimilation (6), in cell wall (5), osmotic (8), and oxidative (5) stress responses, in metal homeostasis (6), butyrolactone response (4), and others processes (38). Regulators are either known to have a positive (Pos) or negative (Neg) impact on ACT biosynthesis or a predictable one (Neg or Pos within brackets followed by a question mark) or an unknown one (question mark). Regulators in black are those that were more abundant in SC than in *SL*, whereas regulators in red are those that were more abundant in *SL* than in *SC*. The numbers within brackets are referring to publications demonstrating the positive or negative regulatory role played by each transcriptional regulator in the regulation of ACT biosynthesis. Regulators in bold are those that were shown to interact directly with the promoter region of actII-ORF4 which encodes the specific activator of the ACT cluster. (SigB/SCO0600) is within brackets since it is known to be involved in osmotic stress response and to have impact on ACT biosynthesis, but it was not detected in our study.

List of Transcriptionnal Regulators and Regulatory Protein Kinases	Impact on Act Biosynth
**Regulation of Phosphate and/or Nitrogen up-take and assimilation**	
PhoR/SCO4229-PhoP/SCO4230	Neg [77,78]
**MtrA/SCO3013**	Neg [181]
GlnR/SCO4159	Neg [86]
DasR/SCO5231 (Regulation of N-acetylglucosamine metabolism)	?
NnaR/SCO2958 (Regu of nitrate-nitrite up-take and assimilation)	(Neg?)
SCO7699/EshA (Stringent response)	Pos [159]
**Cell wall stress**	
SigE/SCO3356	Neg [42,44]
SigU/ SCO2954	?
TCS CseB/CseC (SCO3358/SCO3359)	Neg [42]
TCS VanR/VanS	?
HypR/SCO6294 (?)	?
	?
**Osmotic stress**	
(SigB^ND^/SCO0600)	(Neg?)
SigH/SCO5243	?
SCO5244 (anti-sigma factor H)	?
BldG/SCO3549 (anti-anti-sigma factor)	Pos [36]
TCS OhkA/OrrA (SCO1596/SCO3008)	Neg [94,95]
TCS OsaA/SCO5748 -OsaB/SCO5749	Neg [61]
Prst/SCO3066 (Anti sigma I factor)	?
**Oxidative Stress**	
WblA/SCO3579	Neg [101]
SigR/SCO5216	(Neg ?)
SCO2647 (MarR-like regulator)	(Neg?)
SCO4180 (Fur-like regulator)	(Neg?)
SigN/SCO4034 (general stress response)	Pos [49,50]
**Metal homeostasis**	
TCS CutR/SCO5862-CutS/SCO5863 (copper homeostasis)	Neg [104]
TCS RapA1/SCO5403- RapA2/SCO5404 (copper homeostasis?)	Pos [110]
SCO0487 (regulator of the coelibactin biosynthetic cluster/zinc homeostasis)	?
DmdR1/SCO4394 (regulator of the desferrioxamine biosynthetic cluster/iron homeostasis)	?
AbsC/SCO5405 (MarR-like regulator/zinc homeostasis)	Neg [183]
Zur /SCO2508 (zinc homeostasis)	Neg [141]
**Butyrolactone**	
CprB/SCO0671 (γ-butyrolactone-responsive regulator)	Neg [162,163]
SlbR/SCO0608 (γ-butyrolactone-responsive regulator)	Neg [113]
ScbA/SCO6266 (gamma-butyrolactone synthesis)	Neg [56,57]
AfsR/SCO4426 (γ-butyrolactone-responsive regulator)	Pos [152,156,157]
**Others**	
BldD/SCO5723 (c-di-GMP signaling)	Neg [123]
SCO6808	Neg [115]
SCO1728	Neg [85]
PkaF/SCO2110 (ESTPK)	Neg [155]
RpoZ/SCO1478 (omega sub-unit of RNA polymerase)	Pos [209]
**OdsR**/OdsK-SCO0204/SCO0203 (TCS dormancy/survival regulon)	Pos [91]
**Crp/SCO3571 (Cyclic AMP receptor protein)**	Pos [82]
**AtrA/SCO4118**	Pos [84]
AbrC3/SCO4596 (TCS response regulator)	Pos [71,72,73]
AbrC1/ SCO4598 (TCS sensory histidine kinase)	Pos [71]
SCO6992	Pos [102]
**SCO3932**	Pos [127]
Spa/SCO7629 (starvation sensing protein)	Pos [117]
**MacR/SCO2120** (TCS regulator)	Pos [109]
BldC/SCO4091	Pos [111,112]
TCS **DraR**/Dra K-SCO3063/SCO3062	Pos [62]
Rok7B7/SCO6008 (regulator of the xylose transport operon *xylFGH)*	Pos [105,106]
SCO5785 (TCS response regulator)	Pos [128]
AfsK/SCO4423 (ESTPK)	Pos [195]
Rex/SCO3320 (Sensor of NAD+/NADH poise)	?
SCO5556 (HU DNA-binding protein)	?
LexA/SCO5803 (SOS regulatory protein)	?
OrnA/SCO2793 (Oligoribonuclease)	?
SCO3209 (regulator of para-hydroxybenzoate catabolism)	?
SCO1614 (regulator of the ethanolamine utilization pathway)	?
PkaH/SCO4775 (ESTPK, regulation of sporulation)	?
PglW/SCO6626 (phage resistance)	
SCO1678	Pos [161]
ScrI/SCO4441	Pos [158]
CprB/SCO6071 (regulator of γ-butyrolactone biosynthesis)	Neg [162,163]
NsdA/SCO5582 (Nitrogen metabolism?)	Neg [167,168]
NsdB/SCO7252	Neg [164]
CabB/SCO5464 (calcium homeostasis)	?
SCO2832 (regulator of probable amino acid ABC transporter protein)	?
BldH/SCO2792 (Activator of aerial mycelium formation)	?
SCO2775	?
SCO5804/NrdR (Regulator of ribonucleotide reductase)	?
RamC/SCO6681 (ESTPK, activator of aerial mycelium formation)	?

## 4. Materials and Methods

### 4.1. Bacterial Strains, Media and Culture Conditions

Spores of *S. coelicolor* M145 [6] and *S. lividans* TK24 [7] were prepared from solid Soya Flour Mannitol (SFM) medium [210]. The two strains were grown, in quadruplets, on solid modified R2YE medium, with no sucrose added, on 5 cm diameter Petri dishes. The modified R2YE medium [211] was supplemented with glucose (50 mM) as major carbon source and was either limited (1 mM, no K2HPO4 added) or proficient (4 mM K2HPO4 added) in Pi. Spores (106) of the strains were plated on the surface of cellophane disks (Focus Packaging and Design Ltd., Louth, UK) laid down on the top of agar plates and incubated at 28 °C in darkness for 48 h or 60 h. The time points of 48 h and 60 h were chosen as they correspond to the beginning of the production of the blue pigmented polyketide antibiotic, actinorhodin (ACT), in SC [6]. Mycelial lawns of the four independent biological replicates of each strain were collected with a spatula, washed twice with deionized water, lyophilized, and weighted.

### 4.2. Total Proteins Extraction and Digestion

A volume containing 1μg of protein was injected for each of the 32 samples (2 strains × 2 media × 2 culture times × 4 biological replicates) and subjected to an in-depth shotgun label-free analysis. The samples were alkylated before digestion by Lysyl-Endopeptidase (Wako Chemicals, Richmond, VA, USA) and sequencing-grade-modified trypsin (Promega, Charbonnieres-les-Bains, France). The resulting proteolytic peptides were pre-cleaned, concentrated under vacuum, and stored before mass spectrometry analysis as described previously [8]. Trypsin-generated peptides were analyzed by nanoLC-MSMS using a nanoElute liquid chromatography system (Bruker, Billerica, MA, USA) coupled to a timsTOF pro mass spectrometer (Bruker, Billerica, MA, USA). Then, 1 µg of protein digest in 2 µL of loading buffer (2% acetonitrile and 0.05% trifluoroacetic acid in water) were loaded on an Aurora analytical column (ION OPTIK, 25 cm × 75 µm, C18, 1.6 µm, Bruker, France SAS) and eluted with a gradient of 0–35% of solvent B for 100 min. Solvent A was 0.1% formic acid and 2% acetonitrile in water, and solvent B was 99.9% acetonitrile with 0.1% formic acid. MS and MS/MS spectra were recorded from *m*/*z* 100 to 1700 with a mobility scan range from 0.6 to 1.5 V·s/cm^2^. MS/MS spectra were acquired with the PASEF (parallel accumulation—serial fragmentation (PASEF)) ion mobility-based acquisition mode using a number of PASEF MS/MS scans set as 10. MS and MSMS raw data were processed and converted into mgf files with DataAnalysis software (Bruker, Billerica, MA, USA).

### 4.3. Liquid Chromatography Tandem Mass Spectrometry Analysis

Proteolytic peptides were analyzed by nanoLC-MS/MS (liquid chromatography tandem mass spectrometry) using a nanoElute liquid chromatography system (Bruker) coupled to a timsTOF Pro mass spectrometer (Bruker). Protein digests (1 μg in 2 μL of 2% acetonitrile and 0.05% trifluoroacetic acid in water loading buffer) were loaded on an Aurora analytical column (ION OPTIK, 25 cm × 75 μm, C18, 1.6 μm) and eluted with a gradient of 0–35% of solvent B for 100 min. Solvent A was 0.1% formic acid and 2% acetonitrile in water, and solvent B was 0.1% formic acid and 99.9% acetonitrile. MS and MS/MS spectra were recorded, and raw data were processed and converted into mgf files as described previously [212].

### 4.4. Protein Identifications

Protein identifications were performed against *SC* and *SL* protein database from UniprotKB (15012020) using the MASCOT search engine (Matrix Science, London, UK). Database searches were performed using the following parameters: specific trypsin digestion with two possible miscleavages; carbamidomethylation of cysteines as fixed modification and oxidation of methionines as variable modification. Peptide and fragment tolerances were 25 ppm and 0.05 Da, respectively. Proteins were validated when identified with at least two unique peptides in at least one replicate. Only ions identified with a score higher than the identity threshold, and a false-positive discovery rate of less than 1% (Mascot decoy option) were considered.

### 4.5. Label-Free Mass Spectroscopy-Based Relative Protein Quantification

Protein abundance changes were determined using two label-free mass spectrometry-based quantification methods: spectral count (SC) or MS1 ion intensities named XIC (for extracted ion current). For spectral counting, total MS/MS SC values were extracted from Scaffold software (version Scaffold_4.11.1, Proteome software Inc., Portland, OR, USA) and filtered with 95% probability and 1% false discover rate (FDR) for protein and peptide thresholds, respectively. For MS1 ion intensity, MS raw files were analyzed with Maxquant software (v 1.6.10.43) using the maxLFQ algorithm with default settings and 4D feature alignment corresponding to a match between run function including collisional cross sections (CCS) alignment. Normalization was set as default. Identifications with Andromeda were performed using the same search parameters as those described previously for MASCOT searches. Maxquant software (v 1.6.10.43) can be downloaded at https://www.maxquant.org (MaxPlack Insitute of Biochemistry, Martinsried, Germany)

### 4.6. Protein Abundance Changes and Statistical Analysis

Statistical quantitative analyses were based on two different generalized linear models depending on the type of quantification method used and data that were generated: either spectral counting for a rough relative quantitative protein quantification or XIC from MS1 ion intensities for more accurate and sensitive relative quantifications of low abundant proteins or small abundant changes. The discrete SC (1) and continuous XIC (2) abundances values were modeled, respectively, as follows, and as described previously [212].
SC = μ + strain + medium + time + replicate + strain × medium + strain × time + medium × time + strain × medium × time + ε∼ Pois(λ)(1)
Log2(LFQ) = μ + strain + medium + time + replicate + strain × medium + strain × time + medium × time + strain × medium × time + ε∼ N(0,σ).(2)

Terms represent fixed effect of the different conditions and their interactions for each protein abundance. The residual error (ε) follows a Poisson distribution [Pois(λ)] or a normal distribution for SC and Log2 (LFQ), respectively. Effects were estimated by maximum likelihood. Statistical significances were calculated using likelihood ratio tests based on the analysis of deviance. *p*-values were adjusted using the Benjamini–Hochberg procedure for multiple testing correction. For each protein, first, a significant difference in abundance for the 66 combinations of pairwise comparisons was set at 10 for SC and 1 Log2 fold change for LFQ values. Second, a threshold of 10 significant pairs (minimum number of significant pairwise comparisons expected for a differential protein abundance in at least two conditions) and an adjusted *p*-value of 0.05 was used to consider a protein abundance as significantly variable. These statistical analyses were performed with a homemade R script derived from R Studio (version 1.4.17.17) downloaded at https://www.rstudio.com/products/rstudio/older-versions/, accessed on 14 November 2022 and using the described models and parameters. A threshold of five significant pairs (minimum number of significant pairwise comparisons expected for a differential protein abundance in at least two conditions) was used to consider a protein abundance as significantly variable. All the statistical analyses were performed by a homemade R script and are provided I Appendix A.

## Data Availability

Our mass spectrometry proteomics data have been deposited to the ProteomeXchange [209] Consortium via the PRIDE [213] partner repository with the dataset identifier PXD029263 and 10.6019/PXD029263.

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
