# Peer review of "Comparative Proteomic Analysis of Transcriptional and Regulatory Proteins Abundances in S. lividans and S. coelicolor Suggests a Link between Various Stresses and Antibiotic Production"

_ijms, 2022, doi:10.3390/ijms232314792_

Round 1

Reviewer 1 Report

The manuscript “Comparative analysis of the regulome of S. lividans and S. coelicolor suggest a link between various stresses and antibiotic production” shows a bioinformatics analysis of the differences between the S. coelicolor and S. lividans proteomes focusing on transcriptional proteins (RNA polymerases, sigma and antisigma factors, antiterminators, etc.) regulatory proteins (transcriptional regulators, sensory histidine kinases and eukaryotic-like serine/Thr kinases). Despite that the proteomic data were from a recent manuscript of the same group, the data mining approach of this manuscript focused on the regulatory proteins, instead of energy metabolism (ATP production) analysed in the previous manuscript. This new manuscript reveals interesting new knowledge about regulatory proteins and stresses modulating secondary metabolism activation in S. coelicolor.

Some points can be considered prior to publication:

1.      Quantitative data and statistical parameters. Authors made statistical analyses, but these data are not shown. Can authors provide a supplementary table indicating the abundance values (in de different biological replicates and the average) and q-values (i.e. statistical significance of the variations) of all the proteins shown in the heatmaps? This information will be very useful for authors interested in the proteins identified in this work.

2.      Title. I suggest to modify the title to indicate that results are from a proteomic work (as it is written is not possible to know if it is proteomics or transcriptomics). I also suggest to discard the term regulome in the title because is difficult to know what is regulome without further explanation. The title might be something like: Comparative analysis of transcriptional and regulatory protein abundances in S. lividans and S. coelicolor …”

3.      Abstract, line 19. Again, the term regulome in the abstract is difficult to understand without further explanation. I suggest change it by “regulatory proteins (transcriptional regulators, sensory histidine kinases and eukaryotic-lik serine/Thr kinases)”

4.      Line 76. What is “transcriptional apparatus”, I guess that it refers to RNA polymerases and transcriptional regulators. Please, clarify. I suggest to discard the term “transcriptional apparatus” and describe the proteins (RNA polymerases, transcriptional regulators, etc.).

5.      Line 93. “Transcriptional apparatus”. See previous comment. Perhaps it can be changed by “transcriptional proteins”. Figure 1; why SLIV04800, SCO5216, SCO2300, SCO1490, SLIV_1608, SLIV_24680 belong to the transcriptional apparatus?

6.      Line 641. 358 proteins. What are these proteins? Are they the sum of transcriptional apparatus + transcriptional regulators?

Author Response

We first wish to thank the  reviewer for his/her careful reading of our manuscript, for his/her wise comments and suggestions of modifications and additions (Table S1) to the manuscript. All changes recommended are highlighted in yellow in the revised version 1 of the manuscript.

  1. Quantitative data and statistical parameters. Authors made statistical analyses, but these data are not shown. Can authors provide a supplementary table indicating the abundance values (in de different biological replicates and the average) and q-values (i.e. statistical significance of the variations) of all the proteins shown in the heatmaps? This information will be very useful for authors interested in the proteins identified in this work.

The requested Table S1 is now provided.

  1. I suggest to modify the title to indicate that results are from a proteomic work (as it is written is not possible to know if it is proteomics or transcriptomics). I also suggest to discard the term regulome in the title because is difficult to know what is regulome without further explanation. The title might be something like: Comparative analysis of transcriptional and regulatory protein abundances in S. lividans and S. coelicolor …”

The new title is “Comparative proteomic analysis of transcriptional and regulatory proteins abundances in S. lividans and S. coelicolor suggests a link between various stresses and antibiotic production.

  1. Abstract, line 19 Again, the term regulome in the abstract is difficult to understand without further explanation. I suggest change it by “regulatory proteins (transcriptional regulators, sensory histidine kinases and eukaryotic-lik serine/Thr kinases)”

This was corrected.

  1. Line 76 What is “transcriptional apparatus”, I guess that it refers to RNA polymerases and transcriptional regulators. Please, clarify. I suggest to discard the term “transcriptional apparatus” and describe the proteins (RNA polymerases, transcriptional regulators, etc.).

The new sentence is now “The present study indicated that the abundance of the vast majority of proteins of the transcriptional apparatus, of one or two component transcriptional regulators and of regulatory proteins such  eukaryotic-like serine/threonine protein kinases, common to SC and SL, greatly differed between the 2 strains and responded differently to Pi availability.

  1. Line 93/. “Transcriptional apparatus”. See previous comment. Perhaps it can be changed by “transcriptional proteins”.

This was changed.

  1. Figure 1; why SLIV04800, SCO5216, SCO2300, SCO1490, SLIV_1608, SLIV_24680 belong to the transcriptional apparatus?

We think that the 3 SCO could be considered as belonging to the transcriptional apparatus but not the 3 SLIV so the latter were removed from the heat map of Figure 1 and a new heat map is provided in Figure 1..

SCO5216 is sigR, an RNA polymerase sigma factor involved in the resistance to oxidative stress.

SCO2300 is annotated as a predicted nucleic acid-binding protein since it contains a Zn-ribbon domain.

SCO1490 bears similarities with the anti-termination factor NusB

SLIV_16085 putative 2'-5'-oligoadenylate synthetase 1-like protein.

SLIV_04800, putative ribonuclease

SLIV_24680, putative ribonuclease E/G

  1. Line 641. 358 proteins. What are these proteins? Are they the sum of transcriptional apparatus + transcriptional regulators?

In the new version of the text we have 42-3 (SLIV) = 39 components of the transcriptional apparatus + 293 one or two components transcriptional regulators + 23 eukaryotic-like protein kinases, so a total of 355 regulatory proteins.  

Reviewer 2 Report

This manuscript represents part two of a proteomic study comparing two closely related antibiotic producers, Streptomyces coelicolor A3(2) and S. lividans under phosphate limiting and phosphate proficient conditions. In part I (Lejeune et al., 2022), the authors showed that abundance of proteins of several pathways differed significantly between S. coelicolor A3(2) and S. lividans. These pathways included respiration and central carbon metabolism suggesting oxidative stress to be a trigger of Actinorhodin production in S. coelicolor A3(2).

In the “part II” manuscript the same proteomic data are used to compare the abundance of proteins of the transcriptional apparatus and its regulation. This study revealed that S. coelicolor A3(2) suffers from nitrogen stress, as well as cell-wall and osmotic stress resulting in increased Actinorhodin production in comparison with S. lividans.

The manuscript is difficult to evaluate, since the results section only consists of a list of S. coelicolor and S. lividans proteins which are differentially synthesized in dependence of phosphate availability. Nevertheless, the data revealing the intertwining of the various stress response systems and the variations in two closely related Streptomyces species are highly interesting. The conclusions drawn in the discussion sections are valid and the hypotheses plausible, and in many cases supported by experimental evidence from other (published) studies.

I only have minor comments.

1.    Language: The use of the term protein/gene is not adequate at many positions in the manuscript.

·         (Expression of) A gene is upregulated, not a protein!  (eg. line 281, SCO4158 might regulate GlnR (protein!) expression.

·         A protein does not locate downstream of a protein (e.g. line 277, LacI regulator SCO4158…..downstream of GlnR/SCO4159).

2.    Gene names and species should be written in italics (throughout the whole manuscript)

3.    Fig. 10 shows the abundance of serine/threonine protein kinases. The antagonists of the kinases, the phosphatases should be of similar importance. Were the serine/threonine phosphatases excluded from this analysis?

4.    Line 433. The korSA like regulatory gene is part of an pSAM2-like ICE element (and not upstream of prophage like genes.

Some typing errors:

Line 61. The study of…

Line 136. The expression of numerous…..

Line 329. (Cluster C)..

Line 389. (Cluster D)..

Line 462. Involved

Line 625. In_Pi

Line 745. avaialbility

Line 753. transcriptionnal

Line 925. Formatting error in reference section causes shift in numbering

Author Response

We first wish to thank the reviewer for his/her careful reading of our manuscript, for his/her wise comments and suggestions of modifications and additions (Table S1) to the manuscript.
All changes recommended are highlighted in yellow in the revised version 1 of the manuscript.

  1.  

    Language: The use of the term protein/gene is not adequate at many positions in the manuscript.
  • (Expression of) A gene is upregulated, not a protein! (eg. line 281, SCO4158 might regulate GlnR (protein!) expression.
  • A protein does not locate downstream of a protein (e.g. line 277, LacI regulator SCO4158…..downstream of GlnR/SCO4159).

Corrections were made throughout the manuscript.

  1. Gene names and species should be written in italics (throughout the whole manuscript)

Corrections were made throughout the manuscript.

  1. Fig. 10 shows the abundance of serine/threonine protein kinases. The antagonists of the kinases, the phosphatases should be of similar importance. Were the serine/threonine phosphatases excluded from this analysis?

We did an heat map corresponding to Ser/Thr protein phosphatases (see attached file) but since these proteins are not as well defined as Ser/Thr protein kinases and show complex abundance pattern, we have decided to omit this heat map. We can still include it in Supplementary data if the reviewer feels strongly about it.

  1. Line 433. The korSA like regulatory gene is part of an pSAM2-like ICE element (and not upstream of prophage like genes.

Correction was done.

Some typing errors:

Line 61. The study of…

Line 136. The expression of numerous…..

Line 329. (Cluster C)..

Line 389. (Cluster D)..

Line 462. Involved

Line 625. In_Pi

Line 745. avaialbility

Line 753. transcriptionnal

Line 925. Formatting error in reference section causes shift in numbering

Corrections were done.
